# Predicted Environmental Risk Assessment of Antimicrobials with Increased Consumption in Portugal during the COVID-19 Pandemic; The Groundwork for the Forthcoming Water Quality Survey

**DOI:** 10.3390/antibiotics12040652

**Published:** 2023-03-25

**Authors:** Anabela Almeida, Cristina De Mello-Sampayo, Ana Lopes, Rita Carvalho da Silva, Paula Viana, Leonor Meisel

**Affiliations:** 1Centro de Investigação Vasco da Gama (CIVG), Departamento de Ciências Veterinárias, Escola Universitária Vasco da Gama (EUVG), Campus Universitário de Lordemão, 3020-210 Coimbra, Portugal; 2Coimbra Institute for Biomedical Imaging and Translational Research (CIBIT), Universidade de Coimbra, 3000-548 Coimbra, Portugal; 3Laboratory of Neuroinflammation, Signaling and Neuroregeneration, Research Institute for Medicines (iMed.ULisboa), Faculty of Pharmacy, Universidade de Lisboa, 1649-003 Lisbon, Portugal; 4Agência Portuguesa do Ambiente (APA), Rua da Murgueira, 9, 2610-124 Amadora, Portugal; 5Research Institute for Medicines (iMed.ULisboa), Faculty of Pharmacy, Universidade de Lisboa, 1649-003 Lisbon, Portugal; 6Biosafety Unit, Instituto Gulbenkian de Ciência, 2780-156 Oeiras, Portugal; 7Laboratory of Systems Integration Pharmacology, Clinical and Regulatory Science, Research Institute for Medicines (iMED.Ulisboa), 1600-277 Lisbon, Portugal

**Keywords:** antimicrobials, antimalarials, antivirals, consumption, COVID-19, environmental risk, water survey

## Abstract

The environmental release of antimicrobial pharmaceuticals is an imminent threat due to ecological impacts and microbial resistance phenomena. The recent COVID-19 outbreak will likely lead to greater loads of antimicrobials in the environment. Thus, identifying the most used antimicrobials likely to pose environmental risks would be valuable. For that, the ambulatory and hospital consumption patterns of antimicrobials in Portugal during the COVID-19 pandemic (2020–2021) were compared with those of 2019. A predicted risk assessment screening approach based on exposure and hazard in the surface water was conducted, combining consumption, excretion rates, and ecotoxicological/microbiological endpoints in five different regions of Portugal. Among the 22 selected substances, only rifaximin and atovaquone demonstrated predicted potential ecotoxicological risks for aquatic organisms. Flucloxacillin, piperacillin, tazobactam, meropenem, ceftriaxone, fosfomycin, and metronidazole showed the most significant potential for antibiotic resistance in all analysed regions. Regarding the current screening approach and the lack of environmental data, it is advisable to consider rifaximin and atovaquone in subsequent water quality surveys. These results might support the forthcoming monitorisation of surface water quality in a post-pandemic survey.

## 1. Introduction

The environmental release of antimicrobials is of considerable concern due to the potential ecosystem impacts and the development and spread of resistance.

Antimicrobials comprise medicines used to treat infections, and they include antibiotics, antivirals, antifungals and antiparasitics. This pharmacotherapeutic group mainly contains antibacterial compounds and fewer antifungal, antiviral or antimalarial drugs [1].

Currently, the clinical transversal constraints concerning these medicines are the scarcity of new molecules in development and their antimicrobial resistance (AMR), which occurs when bacteria, viruses, fungi and parasites no longer respond to a drug, increasing the risk of disease spread [2,3,4]. For instance, bacteria resistant to the antibiotic colistin have been detected in various regions. This antibiotic is the only last-resort treatment for life-threatening infections caused by the carbapenem-resistant Enterobacteriaceae. According to the World Health Organization (WHO), in 2018, about half a million new cases of rifampicin-resistant tuberculosis (RR-TB) were globally identified [1]. The prevalence of drug-resistant viral, fungal and parasitic infections is also increasing; therefore, for the long-term treatment of chronic infections (e.g., HIV and malaria), efficient therapy is commonly achieved with combination therapy [5,6].

Antimicrobials are not completely metabolised in the body; they are excreted in the urine and the faecal matter as parent compounds and metabolites, pharmacologically active or not, undergoing incomplete removal in conventional wastewater treatment [7]. Consequently, different amounts of such drugs and their metabolites reach aquatic systems with potential ecotoxicological effects on wildlife and AMR onset.

Previous research has focused on antibiotic detection in aquatic systems, including surface and groundwater systems [8,9,10], as well as on identifying resistance genes and resistant bacteria in various environmental compartments [11,12]. However, evidence regarding antiviral and antimalarial medicines’ environmental occurrence and behaviour is scarce and covers only limited geographical regions [5].

Recently, the world underwent a severe and challenging period due to the COVID-19 outbreak; consequently, the consumption of antimicrobials is expected to increase [13,14,15], likely leading to higher loads in the environment. Therefore, it would be valuable to identify the antimicrobials that are most used and likely to pose the most significant environmental risks. In this context, a predicted risk assessment screening approach based on exposure and hazard could be used to support the forthcoming gathering of monitorisation data for a quality survey of post-pandemic surface water. For this purpose, we evaluated the consumption patterns of antimicrobials in Portugal during the emergent COVID-19 pandemic period (2020–2021) dispensed in ambulatory care and used in hospitals, and we compared them with those of 2019. Further, we identified the antimicrobials that pose potential environmental risks by combining consumption, excretion rate, ecotoxic endpoint, and microbiological endpoint data in five different regions of Portugal.

## 2. Results

### 2.1. Antimicrobial Consumption—Hospital and Ambulatory Sectors

Over the study period from January 2019 to December 2021, antibiotics comprised most of consumed antimicrobials (80.6%–212.7 t), followed by antivirals (15.2%–40.2 t), antimalarials (2.4%–6.3 t) and antifungals (1.8%–4.8 t).

Besides antimalarials, the pharmacotherapeutic groups demonstrated reductions in their consumption levels during 2020–2021 (COVID-19 pandemic time) compared with 2019 for both the hospital and ambulatory sectors.

The amount of antimicrobials dispensed during ambulatory care during the study period was 193.6 t, about 2.74 times the amount used in hospitals. Regarding the pharmacotherapeutic groups, antivirals were consumed primarily in hospitals (31.5 t versus 8.7 t in ambulatory care); in contrast, antibiotics, antimalarials and antifungals accounted for 174.7, 5.8, and 4.4 t dispensed in ambulatory care and 38.0, 0.56, and 0.45 t used in hospitals, respectively. Comparing the years of 2020 and 2021 to 2019 revealed decreases in antibiotic, antiviral, and antifungal consumption in ambulatory and hospital use. Nevertheless, in hospitals, antibiotic use approached their 2019 values in 2021, while for antivirals, a 2019-like consumption trend was seen in 2020 (Figure 1).

Except for antimalarials, negative trends from −2.5% (hospital antivirals) to −15.0% (hospital antifungals) was observed for the compound annual growth rates (CAGRs) for all antimicrobial groups over the study period. However, this tendency was almost imperceptible for the ambulatory consumption of antivirals (CAGR = −0.56%) and the hospital use of antibiotics (CAGR = −0.023%). Figure 1 shows the antibiotic, antiviral, antimalarial, and antifungal consumption rates in the hospital sector and their CAGRs across the study period.

### 2.2. Consumption Analysis per Therapeutic Class and Its Active Substances

Table 1 illustrates the annual growth rate (AGR) assessment of the active antimicrobial substances during the COVID-19 pandemic (2020 or 2021) and during 2019 used in hospitals and dispensed in ambulatory care.

According to the consumption data obtained for 116 active substances, only 22 out of 26 substances showed increase rates with potential environmental concentrations during the pandemic period (2020 and/or 2021) compared with the pre-covid period (2019). The four substances with consumption amounts of below 1.8 kg and without potential environmental concentrations were the antifungals echinocandins (caspofungin, micafungin, and anidulafungin) and voriconazole. The other 22 drugs were distributed in the following classes—antimicrobials: carbapenems, sulfonamides/trimethoprim, and tetracyclines; antivirals: nucleoside and nucleotide reverse transcriptase inhibitors (NRTIs), integrase inhibitors, nucleosides and nucleotides excl. reverse transcriptase inhibitors (NA excl. RTIs); and antimalarials (see Table 1). Considering their active substances, the major determinant for these results was the enhanced usage of ten compounds (meropenem, trimethoprim, doxycycline, minocycline, emtricitabine, lamivudine, dolutegravir, valaciclovir, hydroxychloroquine and atovaquone). Furthermore, this increased consumption trend was preponderant in the first year of COVID-19 for active antiviral and antimalarial substances and some other antibiotic molecules, namely, metronidazole, rifaximin, fosfomycin and ceftriaxone—the first two mainly used in the ambulatory sector and the other two mainly used in the hospital sector. However, the prevalence of active substances with increased consumption was mainly observed in 2021, as the negative consumption trend observed in 2020 cancelled out most of the active substances.

The antibiotic classes—macrolides (e.g., azithromycin and clarithromycin) and quinolones (e.g., ciprofloxacin)—displayed decreased consumption amounts during COVID-19 relative to 2019, with AGRs of roughly −35.0% and −19.2%, respectively.

The antivirals abacavir, tenofovir, and efavirenz showed consumption amounts above 700 kg. Nevertheless, those values were all lower than those of 2019. The antifungals showed a similar trend. For instance, the most used antifungals of fluconazole, itraconazole, and terbinafine reached average quantities of 200, 178, and 1000 kg, respectively, for 2020–2021, with AGR values in the orders of −32, −13, and −21%, respectively, compared with 2019.

All the active substances included in Table 1 with a positive AGR were selected for analysis in the environmental risk assessment.

### 2.3. Consumption Analysis per Region Based on Fpen Assessment

The consumption data were separately evaluated for the five regions of Portugal. The Lisbon/Tagus Valley, Centre, and North regions are heavily populated, with 3,106,069, 1,663,772, and 3,586,586 inhabitants, respectively. In contrast, the Alentejo and Algarve regions only have 704,533 and 467,343 inhabitants, respectively. Hence, a penetration factor (Fpen)—the proportion of the population treated daily with a specific antimicrobial active substance—was calculated based on consumption per region, defined daily dose values, and the residential population of each area (Equation (3)). The Fpen was only estimated for the active substances listed in Table 1. As a result, regardless of the number of inhabitants, different patterns were obtained for the distinct locales. For instance, the highest proportions of populations treated daily with carbapenems (meropenem), cephalosporins (ceftriaxone and cefazolin), metronidazole and vancomycin were seen in the Centre region, followed by the North region. However, excluding ceftriaxone (Figure 2a), the Lisbon/Tagus Valley region showed the highest proportion of people treated daily with the remaining substances.

Unexpectedly, an enhanced daily use of antimicrobials (mainly antivirals) in the Algarve region was observed even though it has the fewest inhabitants of the studied areas, with 51% less people than the Alentejo region (the second least populated area), which did not show a similar trend (Figure 2c). In contrast, piperacillin/tazobactam seemed to be the only active substances presenting higher Fpen values in the Alentejo region than in the Algarve region (Figure 2b).

On the other hand, the daily use of antimalarials (e.g., hydroxychloroquine) was found to progressively increase from the Alentejo region to the Algarve, Centre, and Lisbon regions; the North region showed a daily use rate between those registered for the Alentejo and Algarve regions despite having the largest population (Figure 2d).

### 2.4. Predictive Environmental Risk Assessment of Selected Antimicrobials

#### 2.4.1. Excretion Factor Values

Excretion factor values are displayed in Appendix A.

Among the 22 selected active substances, only four showed low excretion rates: linezolid (30%), minocycline (43%), trimethoprim (48%), and hydroxychloroquine (25%). When comparing the excretion factors of the other 18 compounds, antivirals showed higher excretion rates, with an average of 92% contrasting with the average of 78% for antibiotics. In addition, the elimination route for most of the substances was found to be renal (e.g., tazobactam—80%; acyclovir—92%), while others were excreted via the bile route and thus appeared in faeces (e.g., rifaximin—97%; atovaquone—≥90%).

It is worth noting that the metabolism and excretion data for these substances are often limited in the literature. Therefore, Summary of Product Characteristics (SPC) information from regulatory authorities was consulted to minimise this constraint.

#### 2.4.2. Predictive Environmental Concentration (PEC) Assessment for the Selected Substances

Predicted effect concentrations (PECs) were assessed using Equation (4) and based on the total (hospital and ambulatory) consumption amount of 2021, which was the year showing the most significant number of substances with positive annual growth rates compared with 2019 (see Table 1). From this perspective, the predictive environmental concentration in surface water was calculated for each region, and these values were refined by considering the different substances’ metabolisms and the proportions of the treated populations.

The analysis of the predictive environmental concentration of active substances in surface water for all regions showed a range of PEC values varying from a maximum of 2.17 µg/L for piperacillin to a minimum of 0.03 µg/L for linezolid. Figure 3 illustrates the characterisation of exposure by region, and Figure 4 shows the total exposure of selected active substances in surface water. The characterisation of exposure disclosed the highest predicted high concentrations of various active substances in the Lisbon/Tagus Valley region, followed by the Centre, North, Algarve, and Alentejo regions (Figure 3). For each region, the substances most representative of and therefore significant for surface water monitorisation were: acyclovir and fosfomycin for all areas (Figure 4), emtricitabine for the Lisbon/Tagus Valley and Algarve regions, lamivudine for the Centre region, and Alentejo and raltegravir for the North region. In addition, the following predicted environmental loads were observed: 6.9 µg/L of antibiotics, 6.5 µg/L of antivirals, and 0.54 µg/L of antimalarials.

#### 2.4.3. Predicted No-Effect Concentration Assessment and Risk Characterisation of Selected Substances

Appendix A provides the predicted no-effect concentrations (PNECs) for the ecotoxicological and microbiological risk assessment. Ecotoxicological risk refers to the assessment of hazard potential regarding the aquatic organisms, and microbiological risk refers to resistance selection risk.

Each calculated PNEC was derived from NOECs/EC_10_/EC_50_ (PNEC_ECOtox_) with sub-MICs (PNEC_subMIC_) for antibiotics. In analysing the active substances, emtricitabine showed the highest PNEC by fish *Pimephales promelas* (PNEC 610 µg/L; Appendix A), followed by cobicistat (484 µg/L) and raltegravir (380 µg/L), while atovaquone (antimalarial) and rifaximin (antibiotic) showed the lowest PNECs (0.83 ng/L by *Ceriodaphnia dubia* and 76 ng/L by *Anabaena* sp., respectively).

The estimated risks calculated for the five studied mainland Portugal regions are reported in Table 2. Among the 22 selected substances that posed predicted potential ecotoxicological risk were rifaximin and atovaquone, with calculated risks for aquatic organisms (RQ_ECOtox_) ranging from 1.05 to 2.11 and from 6.63 to 65.1, respectively, depending on region. However, moderate risk was still observed for cefazolin, ceftriaxone and meropenem. Removing the Alentejo region, which only presented a high risk for rifaximin, the predicted risks in the remaining areas were ranked in ascending order as follows: the Centre, Algarve, Lisbon/Tagus Valley, and North regions.

The microbial risk quotient (RQ_MICsub_) was assessed regarding the predicted lowest minimal inhibitory concentrations for resistance selection concerning each antibiotic and its exposure to surface water. It was found that most substances showed an elevated-to-moderate risk quotient, with impacts on all regions; flucloxacillin (RQ = 0.67–1.6), piperacillin (RQ = 3.0–10.3), tazobactam (RQ = 0.19–0.64), meropenem (RQ = 0.81–2.9), ceftriaxone (RQ = 1.5–5.3), fosfomycin (RQ = 0.11–0.22), metronidazole (RQ = 0.80–2.08) exhibited the most significant potential for antibiotic resistance. Due to their low predicted surface water concentrations, minocycline, vancomycin, and linezolid did not indicate the possibility of causing risk.

## 3. Discussion

It is supposed that the greater the antimicrobial usage, the higher the load expected to reach the environment. Thus, we developed an approach based on antimicrobial consumption for systemic use to identify those active substances with increased usage during 2020 and 2021 (pandemic period) compared with 2019, and the predictive ecotoxicological and microbiological risks were evaluated in five different regions of Portugal.

### 3.1. Antimicrobial Consumption

When we analysed the antimicrobial usage (overall national level), it was readily noticeable that differently from antimalarials, decreases in ambulatory and hospital use were seen for the antibiotic, antiviral, and antifungal groups. These outcomes were evidenced by the negative CAGR trend observed over the study period. Unfortunately, it is impossible to compare our results with previously reported consumption data because these data are often provided in terms of a defined daily dose (DDD), a WHO statistical measure of drug consumption. Nevertheless, our estimated trends provide important insights and can be effectively discussed; for instance, the observed decreased antibiotic use aligns with what was documented for Portugal by the Annual Surveillance Report for 2021 from European Centre for Disease Prevention and Control (ECDC) and by Silva et al. [13,16].

Given the main objective of the present paper, we identified 22 active antimicrobial substances (Table 1) with increased consumption rates during the pandemic that may cause environmental risks. Different pharmaceutical medicine patterns were observed for 2020 and 2021, which may have been related to pandemic evolution. The first year corresponded to the beginning of the pandemic and the first wave of COVID-19. Behavioural interventions such as lockdowns, social distancing, and the usage of masks were then implemented, likely decreasing other infections and access to medical appointments. In addition, recommendations from the WHO narrowed antibiotics use [17,18,19,20,21]. These features are considered possible sources of the perceived significant negative growth rates of macrolides (e.g., azithromycin and clarithromycin) and quinolones (e.g., ciprofloxacin), mostly dispensed in ambulatory care. However, an increasing use trend was observed in 2020, mainly for five antiviral substances (emtricitabine, lamivudine, dolutegravir, raltegravir and cobicistat), four antibiotic substances (ceftriaxone, fosfomycin, metronidazole and rifaximin), and two antimalarial substances (hydroxychloroquine and atovaquone).

Nevertheless, the use of antimicrobials in 2021 cancelled out the negative consumption trend detected in 2020, mainly for the considerable number of active substances used in hospitals, which may have been related to the increased severity of diseases and the need for multiple interventions, especially among intensive care unit patients [22]; even though we do not have data on patients admitted to intensive care, a rise in the number of patients hospitalised in 2021 was reported [23]. Likewise, outpatients with mild symptoms of COVID-19 or other less severe diseases were quarantined at home; this was confirmed by the sequent growth of medicines dispensed in ambulatory care [24]. These data are in line with our data, which demonstrate the increased use of fosfomycin, rifaximin, and tetracyclines in the ambulatory sector during 2021.

Thus, in the present work, the observed increased trend in antimicrobial use seems to be related to repurposed medicines in the context of COVID-19, given that the consumption outcomes matched the higher incidence of COVID-19 infections during the second year of the pandemic. For instance, the used antivirals include substances approved for the treatment of human immunodeficiency virus-1 (HIV-1) (emtricitabine, dolutegravir, and raltegravir), chronic hepatitis B (lamivudine) and herpes (valacyclovir). Furthermore, we observed an enhanced daily use of antivirals in two regions of Portugal (Algarve and Lisboa/Tagus Valley), which may be related to the incidence of HIV-1 in these two regions [25]. Moreover, substances other than commonly applied molecules (e.g., ceftriaxone and meropenem) for community-acquired pneumonia or hospital-acquired/ventilator-associated pneumonia coverage were empirically utilised [26] due to their additional characteristics used to face the great challenge of controlling COVID-19. For instance, fosfomycin demonstrated an immunomodulatory effect on human B cell activation; metronidazole has been shown to decrease the levels of several cytokines, with anti-inflammatory properties; rifaximin seems to have great potential to inhibit the interaction of SARS-CoV-2 with ACE2 (the main human receptor for the entry of SARS-CoV-2 into lower respiratory tract epithelial cells); and tetracyclines present anti-inflammatory, neuroprotective, and even antiviral effects [27,28,29,30,31,32,33].

In this study, antimalarials were the sole therapeutic group that demonstrated evident growth during the study period, particularly the active substances of hydroxychloroquine and atovaquone. Both are used in Portugal for treating malaria due to the country’s connection to African territory. However, during 2020, there was a significant increase in hydroxychloroquine usage (especially in the hospital sector), while in 2021, this trend shifted to atovaquone. Increased ambulatory use was only observed for hydroxychloroquine, but it decreased in the second year of COVID-19. These results suggest that, at least in Portugal, hydroxychloroquine was requested for COVID-19 treatment at the beginning of the pandemic, so it was one of the first medicines to be considered for repurposing. However, hydroxychloroquine’s supposed effectiveness was based on several poorly controlled clinical trials, which led to a lack of use at the late stage of COVID-19, when it was replaced by atovaquone [34,35].

For antifungals, our data showed decreased consumption during 2020–2021. This outcome was unexpected since other authors have reported increased incidences of invasive fungal diseases in COVID-19 patients, with voriconazole the first-line treatment [36]. Nevertheless, we found that the used amount of this antifungal in Portugal displayed a negative rate of −7% for 2020 but a positive rate of 12% for 2021 compared with 2019. Echinocandins additionally had positive consumption rates of 8% and 38% for 2020 and 2021, respectively. However, as their maximum consumption rate was 1.8 kg, they were not considered for the environmental risk assessment because the calculated predicted environmental concentration was very low.

Surprisingly, an enhanced daily use of antimicrobials (mainly antivirals) in the Algarve region was observed even though it has the fewest inhabitants, with 51% less than the Alentejo region (the second least populated area), which did not show a similar trend. The Algarve region, with the lowest resident population, is one of the most touristic areas of Portugal. However, the area’s floating population does not explain the consumption trends observed there, given that tourists were not allowed in Portugal during the lockdown.

### 3.2. Environmental Risk Assessment

Regarding the environmental risks of antimicrobials, two main issues have been raised in the last few decades. First, antimicrobials in surface water may be toxic to aquatic life, producing adverse effects in aquatic ecosystems such as the inhibition of cell proliferation in aquatic organisms, thus affecting their physiology and morphology, as described by Pomati and colleagues [37]. Likewise, antimicrobials can contribute to the spread of AMR even at low or sub-inhibitory concentrations, posing an actual risk to human health [38,39,40].

Considering its adverse effects, some compounds were included in the last watch list (WL), namely, amoxicillin, ciprofloxacin, erythromycin, clarithromycin, azithromycin, sulfamethoxazole, and trimethoprim. Nevertheless, except for trimethoprim, the consumption of the above-mentioned substances did not increase during the pandemic period in Portugal.

Hospital effluent represents a relevant source of antimicrobial residues and antibiotic-resistant bacteria in wastewater [41]. Moreover, the COVID-19 outbreak has further increased hospital waste generation over the past two years due to the high loads of residues in hospital effluent due to an overload of patients. Thus, antimicrobial medicines may have been used on a larger scale. In fact, the present study found a significant increase in these specific hospital-used antimicrobials, demonstrating the role of hospital settings as environmental hotspots. Additionally, our risk assessment results indicated that flucloxacillin, piperacillin, tazobactam, meropenem, cefazolin, ceftriaxone, fosfomycin, metronidazole, and trimethoprim were associated with microbiological risk (higher than 0.1). Piperacillin had the highest RQ (RQ = 10.3, Table 2) in the Lisbon/Tagus region, followed by ceftriaxone and meropenem (RQ = 5.3 and 2.9, Table 2) in the Centre region of Portugal. High proportions of the population treated daily with carbapenems and cephalosporins were also observed in the Centre region of Portugal.

Ceftriaxone has a low potential for bioaccumulation [42]; nevertheless, it is eliminated unaltered in the urine (by glomerular filtration) and bile. Even though ceftriaxone is one of the most prescribed antibiotics in health facilities, there is little information regarding its occurrence in the environment [43]. A recent study in India found high levels of ceftriaxone (from 1.25 to 29.15 µg/mL) in hospital effluent [44].

Tazobactam is a β-lactamase inhibitor used in combination with piperacillin. This molecule presents high renal excretion rates as an unchanged drug (Appendix A) and is considered a very highly mobile hydrophilic substance; therefore, it could move quickly towards surface water. It has already been detected in Portugal [8]. In our study, tazobactam presented an RQMICsub of between 0.19 and 0.64, suggesting a critical microbiological risk.

In the present study, rifaximin and atovaquone were the most concerning substances regarding ecotoxicological effects (RQECOtox > 1). As stated before, the consumption of rifaximin has increased during the pandemic, and its use poses ecotoxicological risks in all considered regions of Portugal. It has a very low absorption rate (<1%) and is almost completely excreted (97%) as the parent compound in faeces, with only a small proportion of the dose excreted in urine (0.32%). Only one metabolite, 25-diacetyl rifaximin, was identified. However, no data on its behaviour on sewage treatment plants or its environmental occurrence were found in the literature; thus, rifaximin should be integrated into the water quality survey approach. In Portugal, rifaximin is authorised for the treatment of acute infectious diarrhoea caused by susceptible microorganisms and the symptomatic treatment of uncomplicated diverticular disease of the colon when associated with dietary fibre therapy [45].

Atovaquone is an antimalarial drug that is primarily eliminated via the liver (more than 90% of the drug excreted in bile is in the parent form), with almost undetectable amounts (0.6%) being eliminated via the kidneys [46,47], contrariwise to hydroxychloroquine that is barely excreted and has a low RQ. The substance’s LogKow = 5.31 at pH 7 predicts a high potential for bioaccumulation [48]. In the present study, atovaquone was highlighted as having a high risk quotient; however, no environmental occurrence data were found in the literature. Thus, atovaquone should also be integrated into the water quality survey approach.

Lately, antiviral drugs have been recognised as emerging contaminants, and their environmental fate and ecotoxicological effects have been investigated [5,49]. Antiviral drugs are among the most predicted hazardous therapeutic classes regarding their toxicity towards algae, daphnids and fish [50]. Thus, knowing the fate, occurrence, and toxicological effects of antiviral drugs in the aquatic environment is mandatory. The incomplete removal of antiviral drugs from effluent may lead to the development of viral resistance with adverse health effects on humans and harmful effects on the environment [51,52]. In contrast to bacteria, viruses are obligate intracellular parasites that rely on host cellular functions to replicate. Thus, most currently available antivirals target specific viral functions, and many have significant toxicities [53]. Although antiviral resistance is rarely mentioned or considered, it should be a point of concern and was addressed by Laughlin and colleagues [53]. According to a recent study, Portugal has the highest systemic use of antiviral drugs in DDD per 1000 inhabitants per day, with HIV drugs presenting the highest percentage of total use [5]. In the present study, no significant increase in consumption was found during the pandemic period. However, an amount of −2.5 CAGR was observed. Nevertheless, data have confirmed the environmental occurrence of antivirals in countries such as Germany [54,55,56], South Africa [57,58], France [59], Kenya [60] and Finland [61]. In addition, abacavir used for HIV infection treatment was detected in 11 of 13 surface sampling stations and one groundwater sampling station in Portugal in a recent study by our research team [8], highlighting the need for antiviral screening programs at the national and global levels.

In particular, acyclovir (ACV) is widely used as an antiherpetic medication, and it is mainly excreted (92%) as an unchanged parent compound [62]. Recent studies indicated the almost complete removal of acyclovir (97%) and lamivudine (>93)% [54] from sewage treatment plants (STPs). Nevertheless, analyses of different environmental samples revealed the presence of transformation products such as the stable carboxy-acyclovir (carboxy-ACV) in surface and drinking water, with concentrations of up to 3200 ng/L and 40 ng/L, respectively [5,63]. Additionally, carboxy-ACV concentrations should not be neglected since *Daphnia magna* toxicity was reported [64]. In our study, we must emphasise that no ecotoxicological risk was observed for the five selected antiviral substances.

Cobicistat was one of the substances that showed an increased consumption rate during the pandemic period in Portugal. Cobicistat is an inhibitor of human CYP3A isoforms [65], and it is used as a pharmacokinetic enhancer that inhibits the metabolism of antiretroviral drugs, boosting protease inhibitors (PIs) and integrase inhibitors to prolong their effects in HIV-infected patients. Cobicistat does not have reported activity against HIV [65]. Regarding ecotoxicological risk, this molecule is not easily biodegradable; however, according to the criteria set by the EU, cobicistat should not be considered a persistent, bioaccumulative, and toxic (PBT) substance, and its use is considered to result in negligible environmental risk [66]. Nevertheless, 86% of this substance is excreted in faeces, and 8% is excreted in urine. Furthermore, this molecule may inhibit the CYP activity of marine species and thereby interfere with the hepatic clearance of xenobiotics from those species. Thus, this molecule may also pose an environmental risk similar to those detected with other known CYP inhibitors belonging to the antifungal group [67]. Nevertheless, no risk was observed with the assessment approach used in the present study.

## 4. Materials and Methods

### 4.1. Consumption of Antimicrobial Drugs

Information regarding medicine consumption data in terms of packaging, pharmaceutical form, and quantitative composition were obtained from the Department for the Medicine’s Economic Assessment of INFARMED, allowing for the estimation of the consumed amount of active substances for each specific antimicrobial drug/year in kg [68].

The collected data refer to the medicines for systemic use in hospitals and the primary health care sector, i.e., medicines dispensed in ambulatory care, within the Portuguese National Health Service. The study period corresponded to the consecutive years of 2020–2021 and the previous year of 2019, which was used to evaluate changes in antimicrobial consumption trends. The consumption data comprised five regions of Portugal’s mainland ranging from the South to the North: Algarve, Alentejo, Lisbon/Tagus Valley, Centre, and North.

Variables were presented as numbers (kg and tons) and percentages. In addition, compound annual growth rates (CAGRs) and annual growth rates (AGRs) were calculated to illustrate changes in the consumption rates of antimicrobials over the three years of the study period and to visualise the growth of antimicrobial consumption for 2020 and 2021 versus 2019.
(1)$$\mathrm{CAGR}\;\left(\%\right)=\left[{\left(\frac{\mathrm{CONEnd}}{\mathrm{CONStart}}\right)}^{\frac{1}{N}}-1\right]\times 100$$

In this equation, CONEnd is the consumption of antimicrobials for the last year, CONStart is the consumption of antimicrobials for the first reported year, and N = 3 is the number of years of reporting (2019–2021).
(2)$$\mathrm{AGR}\;\left(\%\right)=\left(\frac{\mathrm{CONEnd}}{\mathrm{CONStart}}-1\right)\times 100$$
where CONEnd is the consumption of antimicrobials in the last year and CONStart is the consumption of antimicrobials for the year first reported.

The antimicrobial drugs that demonstrated increased consumption—positive annual growth rates—during the pandemic (2020 or/and 2021) compared with the previous year (2019) were selected for environmental risk assessment.

### 4.2. Assessment of Market Penetration Factor (Fpen)

The fraction of a population receiving the active substance (Fpen) was calculated using the following equation [69]:(3)$$\mathrm{Fpen}=\frac{\mathrm{consumption}}{\mathrm{DDD}\;\times \;\mathrm{hab}\;\times 365}$$
where consumption is the amount of antimicrobial drugs used per year (2019–2021), DDD is the defined daily dose [70], hab is the number of inhabitants in Portugal or the different regions of Portugal, and 365 is the number of days per year.

### 4.3. Metabolisation and Excretion Fraction of the Most Representative Antimicrobials

As the parent compound, the proportion of excretion may contribute to a significant or lesser environmental impact related to the corresponding reported consumption [71]. Therefore, excretion fractions were determined by summing the excreted proportion of the active substance (urine and/or faeces) and the ratio of the parent molecule existing as a glucuronide conjugate. It was assumed that all glucuronide conjugates are cleaved to the parent compound in the environment. If no information was given on the nature of the conjugate, we considered the worst-case scenario, wherein all active conjugates were glucuronide conjugates [72]. When more than one final excretion value was given for the unchanged active substance, the worst-case scenario was again considered; thus, the greater one was reported.

### 4.4. Predictive Environmental Risk Assessment of Selected Antimicrobials

A risk-based approach was used to evaluate the possible antimicrobial risk to the surface water compartment. Thus, the ratios between the predicted environmental concentrations (PECs) and the predicted no-effect concentrations (PNECs), either environmental or microbiological, were briefly assessed. It was assumed that a risk quotient (RQ) to the aquatic compartment was indicated when the PEC/PNEC_ECOtox_ ratio was ≥ 1 or PEC/PNEC_MICsub_ ratio was ≥0.1 [73].

We categorised risk as critical if RQ_ECOtox_ ≥ 1 and RQ_MICsub_ ≥ 0.1, moderate for RQ_ECOtox_ between 0.5 ≤ RQ < 1 and RQ_MICsub_ between 0.05 ≤ RQ < 0.1, and no risk for RQ_ECOtox_ < 0.5 and RQ_MICsub_ < 0.05.

#### 4.4.1. Calculation of Predicted Environmental Concentrations in Surface Waters (PECs)

The following formula based on EMA guideline [74] was used to estimate the PEC in the surface water:(4)$$\mathrm{PECsurface}\;\mathrm{water}=\frac{\mathrm{DOSEai}\;\times \;\mathrm{Fpen}\;\times \;\mathrm{Fexcrete}}{\mathrm{WASTEWinhab}\;\times \;\mathrm{DILUTION}}$$
where DOSEai is the maximum daily dose consumed per inhabitant, Fpen is the fraction of a population receiving the active substance (see Section 4.3), Fexcrete is the proportion of the antimicrobial excreted in urine and/or faeces (see Section 4.2), WASTEWinhab is the amount of wastewater per inhabitant per day (133 L inhabit/day [75]), dilution is the dilution from sewage treatment plant (STP) effluent to surface water (default value: 10), and 365 is the number of days per year.

#### 4.4.2. Calculation of Predicted No-Effect Concentrations (PNECs)

As we studied antimicrobials, both environmental predicted no-effect concentration (PNEC_ECOtox_) and PNECs based on minimal inhibitory concentrations (PNEC_MICsub_) were assessed. The estimation was based on the published most sensitive endpoints. Therefore, the PNEC_ECOtox_ values were calculated by using the lowest value of no-observed effect concentration (NOEC), 10% effect concentration (EC_10_), or 50% effect concentration (EC_50_), when the EC10 was not available. Accordingly, an assessment factor was applied [76]. The MICs and minimal selective concentrations were derived from peer-reviewed literature. The PNEC_MICsub_ values were then calculated using an assessment factor to account for the differences between MICs and minimal selective concentrations [12,77].

## 5. Conclusions

Over the study period from January 2019 to December 2021, antibiotics comprised most of the consumed antimicrobials in Portugal, followed by antivirals, antimalarials and antifungals.

Only 22 out of the 116 evaluated active substances showed positive increase rates with potential environmental concentrations during the pandemic period (2020 and/or 2021) compared with the pre-covid period (2019). This increased consumption trend was preponderant in the first year of COVID-19 for active antiviral and antimalarial substances and for some other antibiotic molecules, namely, metronidazole, rifaximin, fosfomycin and ceftriaxone. The prevalence of active substances with increased consumption was mainly observed in 2021.

Considering the proportion of the population treated daily with a specific antimicrobial active substance, different patterns were observed for the distinct Portuguese regions regardless of the number of inhabitants. Higher proportions of people treated daily with carbapenems (meropenem), cephalosporins (ceftriaxone and cefazolin), metronidazole, and vancomycin were seen in the Centre region, followed by the North region. However, the Lisbon/Tagus Valley region showed the highest proportion of people treated daily with the remaining substances. Surprisingly, without a reasonable explanation, an enhanced daily use of antimicrobials (mainly antivirals) in the Algarve region was observed even though it has the fewest inhabitants, with 51% less than the Alentejo region (the second least populated area), which did not show a similar trend. Further attention should be given to the antimicrobial consumption trends of the Algarve region. In contrast, piperacillin/tazobactam was found to be the only active substance, with higher Fpen values in the Alentejo region than in the Algarve region.

Among the 22 selected substances, only rifaximin and atovaquone demonstrated predicted potential ecotoxicological risk for aquatic organisms. Except for the Alentejo region, where only rifaximin presented a high risk, predicted risks were observed in the remaining areas, ranked in ascending order as: the Centre, Algarve, Lisbon/Tagus Valley, and North regions. Moderate risk was still observed for cefazolin, ceftriaxone and meropenem. Considering microbial risk, flucloxacillin, piperacillin, tazobactam, meropenem, ceftriaxone, fosfomycin and metronidazole displayed the most significant potential for antibiotic resistance in all five regions of Portugal. On the other hand, due to their low predicted surface water concentration, minocycline, vancomycin, and linezolid did not present probabilities of causing risk. It turns out that if the PEC was not diluted by 10 (surface water), we could obtain a much higher PEC of the STP effluent (PEC effluent), and then microbiological risk would be relevant, especially for minocycline and doxycycline.

The pattern of increased antimicrobial consumption observed during the pandemic might not be maintained in the post-pandemic period. However, according to the present screening approach and the lack of environmental data, it is advisable to consider rifaximin and atovaquone in the groundwork for the forthcoming water quality survey, as these substances have already reached the environment in hazardous amounts. Furthermore, given the potential of all antibiotics evaluated in the present study to promote antimicrobial resistance and the role of hospitals as hotspots, it is essential to promote the monitoring of positive risk-identified antimicrobials in waste surface water.

## Figures and Tables

**Figure 1 antibiotics-12-00652-f001:**
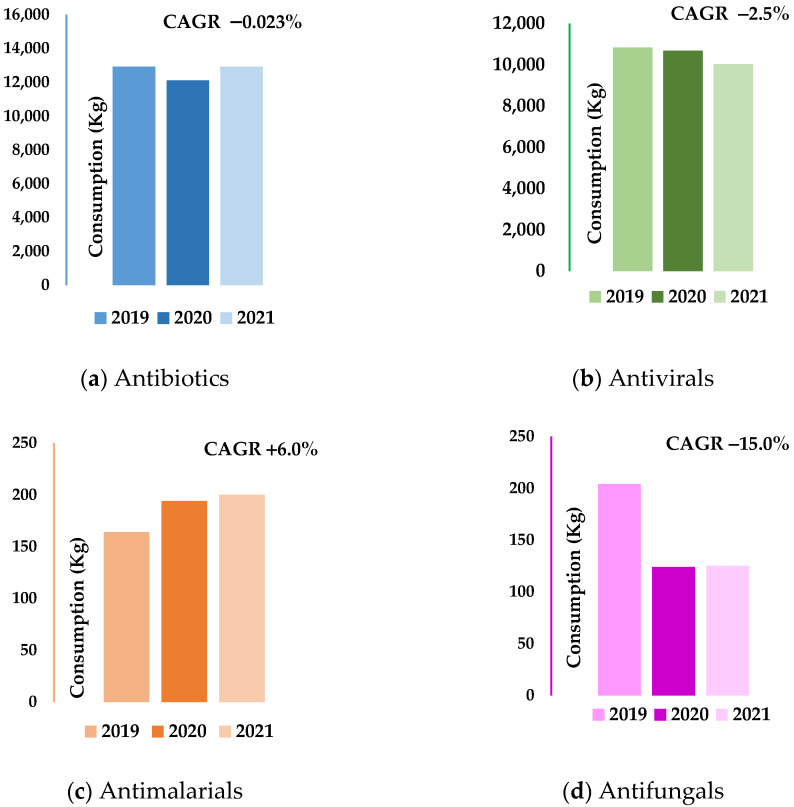
Annual antibiotic, antiviral, antimalarial and antifungal hospital consumption rates (kg). (**a**) Antibiotics—hospital; (**b**) Antivirals—hospital; (**c**) Antimalarials—hospital; (**d**) Antifungals—hospital. CAGR—Compound Annual Growth Rate.

**Figure 2 antibiotics-12-00652-f002:**
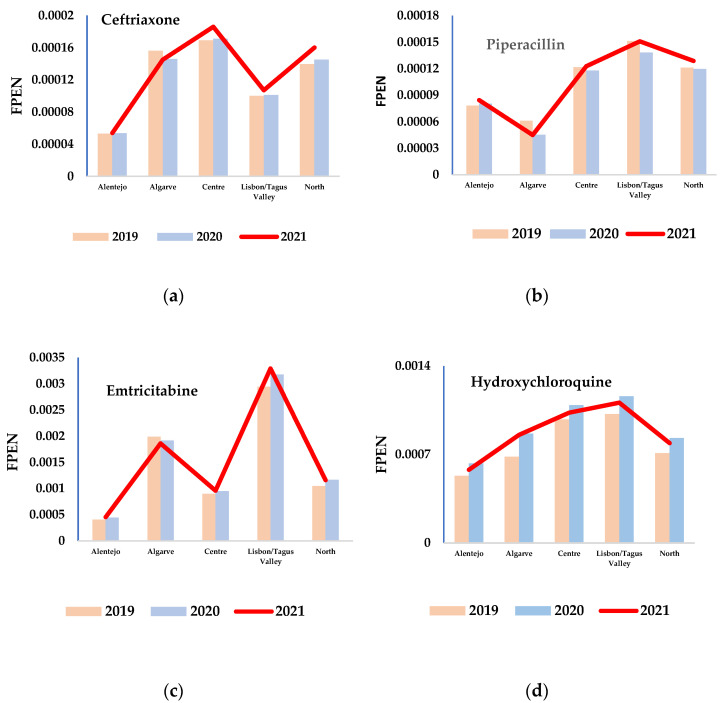
Different trends and patterns of Fpen for four active substances: (**a**) ceftriaxone (antibiotic), (**b**) piperacillin (antibiotic), (**c**) emtricitabine (antiviral), and (**d**) hydroxychloroquine (antimalarial) during the study period (2019–2021).

**Figure 3 antibiotics-12-00652-f003:**
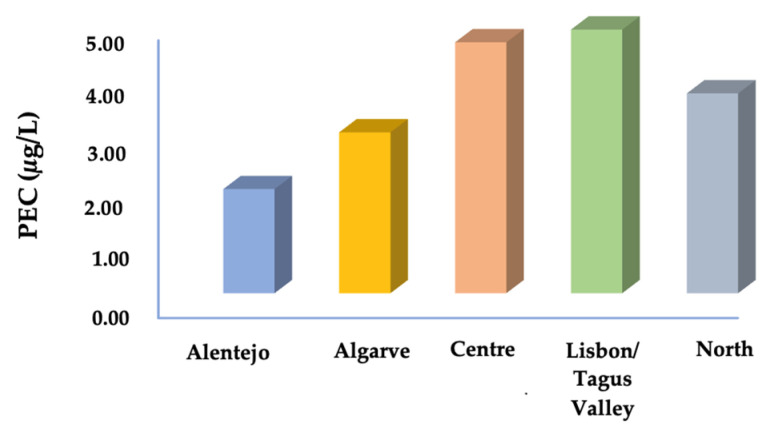
Predicted environmental concentration (exposure) of surface water in different regions. PEC values were obtained from the sum of each PEC-active substance.

**Figure 4 antibiotics-12-00652-f004:**
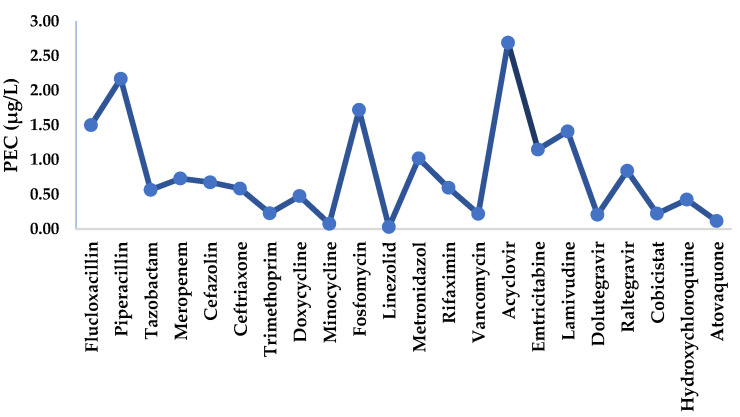
Predicted environmental concentration (exposure) of surface water to selected active substances.

**Table 1 antibiotics-12-00652-t001:** Assessment of AGRs for the antimicrobials’ active substances during the COVID-19 pandemic period (2020 or 2021) compared with 2019 used in hospitals and dispensed in ambulatory care.

Pharmacotherapeutic Group/Class		Antimicrobial Active Substances	2019	2020		2021	
			Control Consumption kg	Consumption kg	AGR %	Consumption kg	AGR %
Antibiotics	Penicillins	Flucloxacillin	283 (H) 2792 (A)	269 (H) 2209 (A)	−4.9 −20.9	326 (H) 1974 (A)	+15.2 −29.3
		Piperacillin	6083 (H)	5787(H)	−4.9	6206 (H)	+2.0
	Enzyme inhibitor	Tazobactam	765 (H)	729 (H)	−4.7	776 (H)	+1.4
	Carbapenems	Meropenem	479 (H)	474 (H)	−1.0	518 (H)	+8.1
	Cephalosporins	Cefazolin	696 (H)	616 (H)	−11.5	716 (H)	+2.9
		Ceftriaxone	879(H)	895 (H)	+1.8	966 (H)	+9.9
	Sulfonamides/ Trimethoprim	Trimethoprim	55 (H) 427 (A)	51 (H) 403 (A)	−7.2 −5.6	52 (H) 447 (A)	−5.4 +4.7
	Tetracyclines	Doxycycline	225 (A)	212 (A)	−5.8	248 (A)	+10.2
		Minocycline	165 (A)	157 (A)	−4.8	179 (A)	+8.5
	Other Antibiotics	Fosfomycin Linezolid Metronidazole	53 (H) 2726 (A) 81 (H) 63 (H) 975 (A)	57 (H) 2735 (A) 79 (H) 54 (H) 830 (A)	+7.5 +0.33 −2.5 −14.3 +14.9	43 (H) 2866(A) 91 (H) 229 (H) 1049 (A)	−19.0 +5.1 +12.3 +263.0 +7.6
		Rifaximin	69 (H) 135 (A)	57 (H) 174 (A)	−17.0 +29.0	60 (H) 274 (A)	+13.0 +102.0
		Vancomycin	306 (H)	305 (H)	−0.3	327 (H)	+6.9
Antivirals	* NRTIs	Emtricitabine	1217 (H)	1312 (H)	+7.8	1340 (H)	+10.1
		Lamivudine	1378 (H)	1459 (H)	+5.8	1511(H)	+9.7
	Integrase inhibitors	Dolutegravir	192 (H)	229 (H)	+19.3	253 (H)	+31.8
		Raltegravir	1321 (H)	1406 (H)	+6.4	1224 (H)	−7.3
	** NA excl. RTIs	Valacyclovir	1405 (H)	1392 (H)	−0.9	1560 (H)	+11.0
	CYP450 3A	Cobicistat	242 (H)	259 (H)	+7.0	235 (H)	−2.9
Antimalarials		Hydroxychloroquine	4.8 (H) 1578 (A)	17.0 (H) 1802 (A)	+45.8 +14.2	5.2 (H) 1727 (A)	+14.6 +9.4
		Atovaquone	158 (H)	176 (H)	+11.4	192 (H)	+21.5

(H)—hospital; (A)—ambulatory; * NRTIs—nucleoside and nucleotide reverse transcriptase inhibitors; ** NA excl. RTIs—nucleosides and nucleotides excl. reverse transcriptase inhibitors.

**Table 2 antibiotics-12-00652-t002:** PEC and ecotoxicological/microbiological risk quotients (RQs) for the selected substances in surface water.

Active Substances	Environmental Risk Assessment														
	Alentejo			Algarve			Centre			Lisbon/Tagus Valley			North		
	PEC µg/L 2021	RISK (PEC/PNEC)		PEC µg/L 2021	RISK (PEC/PNEC)		PEC µg/L 2021	RISK (PEC/PNEC)		PEC µg/L 2021	RISK (PEC/PNEC)		PEC µg/L 2021	RISK (PEC/PNEC)	
		ECOtox	MICsub		ECOtox	MICsub		ECOtox	MICsub		ECOtox	MICsub		ECOtox	MICsub
**Penicillins**															
Flucloxacillin	0.20	0.02	0.67	0.25	0.02	0.83	0.36	0.03	1.2	0.48	0.04	1.6	0.21	0.02	0.70
Piperacillin	0.34	ND	5.7	0.18	ND	3.0	0.50	ND	8.3	0.62	ND	10.3	0.53	ND	8.8
Tazobactam	0.089	0.002	0.36	0.047	0.001	0.19	0.13	0.003	0.52	0.16	0.004	0.64	0.14	0.003	0.56
**Carbapenems**															
Meropenem	0.065	0.22	0.81	0.094	0.31	1.2	0.23	0.77	2.9	0.22	0.73	2.8	0.12	0.4	1.5
**Cephalosporins**															
Cefazolin	0.95	0.40	0.10	0.090	0.38	0.09	0.17	0.71	0.17	0.15	0.63	0.15	0.17	0.71	0.17
Ceftriaxone	0.049	0.15	1.5	0.13	0.39	4.1	0.17	0.51	5.3	0.10	0.30	3.1	0.14	0.42	4.4
**Trimethoprim**															
Trimethoprim	0.020	1.5 × 10^−4^	0.04	0.036	2.7 × 10^−4^	0.07	0.065	4.9 × 10^−4^	0.13	0.06	4.6 × 10^−4^	0.12	0.046	3.5 × 10^−4^	0.092
**Tetracyclines**															
Doxycycline	0.040	0.074	0.020	0.096	0.18	0.05	0.11	0.20	0.06	0.13	0.24	0.07	0.10	0.19	0.05
Minocycline	0.004	0.001	0.004	0.015	0.004	0.015	0.024	0.006	0.024	0.02	0.005	0.02	0.017	0.040	0.017
**Other Antibiotic**															
Fosfomycin	0.21	ND	0.11	0.34	ND	0.17	0.44	ND	0.22	0.41	ND	0.21	0.32	ND	0.16
Linezolid	0.0022	0.002	2.8 × 1^−4^	0.0068	0.008	8.5 × 10^−4^	0.008	0.008	9.5 × 10^−4^	0.008	0.009	0.001	0.004	0.004	5.0 × 10^−4^
Metronidazole	0.10	0.002	0.80	0.21	0.005	1.68	0.26	0.006	2.08	0.26	0.006	2.08	0.19	0.005	1.52
Rifaximin	0.085	1.12	ND	0.080	1.05	ND	0.16	2.11	ND	0.12	1.58	ND	0.15	1.97	ND
Vancomycin	0.011	3.0 × 10^−6^	0.001	0.021	4.8 × 10^−4^	0.003	0.064	0.001	0.008	0.055	0.001	0.007	0.068	0.002	0.009
**Antivirals**															
Acyclovir	0.20	0.008	ND	0.56	0.022	ND	0.74	0.030	ND	0.70	0.028	ND	0.49	0.020	ND
Emtricitabine	0.068	1.1 × 10^−4^	ND	0.28	4.6 × 10^−4^	ND	0.14	2.3 × 10^−4^	ND	0.49	8.0 × 10^−4^	ND	0.17	2.8 × 10^−4^	ND
Lamivudine	0.038	6.1 × 10^−4^	ND	0.13	0.002	ND	0.20	0.008	ND	0.49	0.008	ND	0.27	0.004	ND
Dolutegravir	0.0042	4.4 × 10^−4^	ND	0.024	0.003	ND	0.030	0.007	ND	0.071	0.007	ND	0.036	0.004	ND
Raltegravir	0.040	1.1 × 10^−4^	ND	0.13	3.4 × 10^−4^	ND	0.19	5.0 × 10^−4^	ND	0.27	7.1 × 10^−4^	ND	0.21	5.5 × 10^−4^	ND
Cobicistat	0.0042	8.7 × 10^−6^	ND	0.047	9.7 × 10^−5^	ND	0.014	1.7 × 10^−4^	ND	0.081	1.7 × 10^−4^	ND	0.035	7.2 × 10^−5^	ND
**Antimalarials**															
Hydroxychloroquine	0.056	0.016	ND	0.83	0.024	ND	0.099	0.029	ND	0.11	0.032	ND	0.076	0.022	ND
Atovaquone	0.0002	0.24	ND	0.0055	6.63	ND	0.018	21.7	ND	0.054	65.1	ND	0.038	45.8	ND

ND—data not available; PEC/PNEC = RQ; red for RQ_ECOtox_ ≥ 1 and RQ_MICsub_ ≥ 0.1; yellow for RQ_ECOtox_ between 0.5 ≤ RQ < 1 and RQ_MICsub_ between 0.05 ≤ RQ < 0.1; green for RQ_ECOtox_ < 0.5 and RQ_MIC_ < 0.05.

## Data Availability

Not applicable.

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
