# Peer review of "Predicted Environmental Risk Assessment of Antimicrobials with Increased Consumption in Portugal during the COVID-19 Pandemic; The Groundwork for the Forthcoming Water Quality Survey"

_antibiotics, 2023, doi:10.3390/antibiotics12040652_

Round 1
Reviewer 1 Report
Anabela and coauthors investigated the ambulatory and hospitals consumption patterns of antimicrobials in Portugal during the COVID-19 pandemic (2020-2021) compared with 2019 and conducted an approach to predict and assess risk based on exposure and hazard in the surface water. A total of 22 selected substances with increased consumption were chosen to predict potential ecological risk to aquatic organisms. Rifaximin and atovaquone were verified to have high risk. However, the following comments need to be addressed before publication in Antibiotics.
General comments:
1. I wonder why the daily use of antimicrobials increased in Algarve area with the fewest inhabitants during the pandemic period?
2. Why did you choose the four drugs including ceftriaxone, piperacillin, emtricitabine and hydroxychloroquine as representative antimicrobials among the 22 selected substances? (Figure 2)
3. Why did fungal disease increase while antifungal drug use decrease? Please give a reasonable explanation. (Lines 348-352)
Other comments:
1. Please refer to the figure in the text so that the reader can locate it quickly.
2. Please check the spelling of ‘consumptiom’ and ‘demonstaed’ in line 35 and line 36.
Reviewer 2 Report
Antibiotics, 2023, entitled „Predicted Environmental Risk Assessment of Antimicrobials with increased Consumption in Portugal during the Covid-19 pandemic; The groundwork for the forthcoming water quality survey”.
Almeida et al. give interesting insights in the consumption of antimicrobial, antiviral, antimalarial and antifungal substances at different regions in Portugal during the COVID-pandemic time, and the authors compared the data with the consumptions prior to the COVID outbreak. It was hypothesized, that the outbreak would lead to higher consumption and to higher loads of the substances in the environment. A risk assessment screening approach was carried out with regard to consumption, excretion rates and ecotoxic end-points. Among others, 22 substances had increased consumption but rifaximin and atovaquone had ecological risks. The authors suggest a more detailed investigation with monitoring in further water quality surveys.
The study and the results are interesting to read; however, it became not clear why precisely these two substances came up. A change in the use to the substances was shown based on consumption, but the patient´s illness was not taken into account further. Fewer routine and preventive check-ups were carried out during the pandemic and patients with chronic diseases were treated far less than COVID patients. Therefore, the overall picture of disease also shifted, resulting in a different use of antimicrobials. Therefore, it would be desirable also to consider the number of treatments, diseases and total patient numbers in hospital and outpatient settings. One question is whether the authors now see a decline in substances in the post-COVID pandemic period or whether others are increasing in number again that were already higher in consumption in the years before. Was this increased consumption actually caused by the pandemic? In that case, there would be no need for further investigation to rifaximin and atovaquone. In general, the manuscript is written very well and the results are coherent, however, in my opinion, a higher reflection and consideration of the number of patients in hospitals and in outpatient departments with the corresponding treatments is missing.
Here are few detail remarks; however, the manuscript should be revised for spelling.
- Line 32: …consumption ….
- Line 36: … demonstrated….
- Lines 88 to 70: the sum of the percentages is 100.1 %.
- Line 392: change …Rifaximin… to rifaximin
- Line 446: what means PBT?
Round 2
Reviewer 2 Report
The manuscript has been well revised and the additions and explanations are very helpful for the basic understanding of some assessments and evaluations. The manuscript is linguistically well revised.
Line 346: … displayed a negative rate of -7%....
Line 349: assessment, because…. (comma, but I am not sure)
Author Response
We are very grateful for your comments and suggestions concerning our revised manuscript.
Point 1. Line 346: … displayed a negative rate of -7%....
Response 1: “of” has been added to line 346.
Point 2. Line 349: assessment, because…. (comma, but I am not sure)
Response 2: After careful analysis, we have considered that it is not necessary to add a comma in this situation.